# A Self-Healing Hierarchical Fiber Hydrogel That Mimics ECM Structure

**DOI:** 10.3390/ma13225277

**Published:** 2020-11-21

**Authors:** Kai Li, Yuting Zhu, Qiang Zhang, Xiaoli Shi, Feng Liang, Dong Han

**Affiliations:** 1The State Key Laboratory for Refractories and Metallurgy, Institute of Advanced Materials and Nanotechnology, School of Chemistry and Chemical Engineering, Wuhan University of Science and Technology, Wuhan 430081, China; lik2018@nanoctr.cn; 2Center for Excellence in Nanoscience, National Center for Nanoscience and Technology, Beijing 100190, China; zhuyt@nanoctr.cn (Y.Z.); zhangq@nanoctr.cn (Q.Z.); dhan@nanoctr.cn (D.H.); 3School of Future Technology, University of Chinese Academy of Sciences, Beijing 100049, China

**Keywords:** hydrogel, hierarchy, fiber, self‑healing, wound dressing

## Abstract

Although there have been many studies on using hydrogels as substitutes for natural extracellular matrices (ECMs), hydrogels that mimic the structure and properties of ECM remain a contentious topic in current research. Herein, a hierarchical biomimetic fiber hydrogel was prepared using a simple strategy, with a structure highly similar to that of the ECM. Cell viability experiments showed that the hydrogel not only has good biocompatibility but also promotes cell proliferation and growth. It was also observed that cells adhere to the fibers in the hydrogel, mimicking the state of cells in the ECM. Lastly, through a rat skin wound repair experiment, we demonstrated that this hydrogel has a good effect on promoting rat skin healing. Its high structural similarity to the ECM and good biocompatibility make this hydrogel a good candidate for prospective applications in the field of tissue engineering.

## 1. Introduction

As a typical biological soft matter, the extracellular matrix (ECM) is widespread in various extracellular spaces in the muscles, tendons, and other tissues [1]. The ECM is a hierarchical porous medium composed of fibril bundles (at a micrometer scale) and a filled gel‑like matrix (at a nanometer scale), which keeps organs in place and attaches epithelial tissues to other underlying tissues [2].

The hierarchical structure of the ECM determines its physical characteristics, which ultimately affect the growth and development of organisms [3]. The number of fiber structures and their arrangement within the ECM differ across various tissues and organs; thus, the state of the body and its constituent processes are affected upon changing the state of these fibers [4,5]. For example, excessive fibrosis can occur in the ECM upon excessive deposition of fibrous components (especially collagen), directly leading to an enhancement in ECM stiffness [6,7]. Studies have shown that cells within the ECM attach to fibers and exchange biochemical signals through attachment points [8], such as growth factors that promote cell proliferation through fiber attachment points [9,10]. The nonlinear mechanical properties resulting from an uneven fiber diameter represent another important characteristic of the ECM’s hierarchical structure. For instance, cells in the ECM undergo changes in their phenotype as a function of the force around their fiber attachment points [4,5].

A hydrogel, is a three‑dimensional (3D) polymer network formed by macromolecular chains, and is characterized by extremely high hydrophilicity and widely adjustable physical and chemical properties [11,12,13]. Hydrogels are ubiquitous in nature, with ECM collagen and hyaluronic acid [14] in organisms being examples. In order to simulate their natural counterparts in organisms, researchers have modified the physicochemical and biological properties of hydrogels by integrating chemical and biological activity recognition units (such as molecules that stimulate responses and growth factors) [11,13]. 

At present, biomimetic hydrogels prepared via artificial synthesis have excellent biomimetic properties and application prospects in the fields of drug delivery, tissue engineering, and medical devices [15,16,17]. However, conventional hydrogels still have many limitations in terms of simulating the ECM. First, compared with the complexity of natural tissues, their structure is too simple; second, they lack a stimulus response mechanism to induce cell phenotype changes and the formation of corresponding tissue structures [18,19]. As one example of a biomimetic hydrogel, fiber hydrogels can not only imitate the complex structure of the ECM but also simulate, to a certain extent, its dynamic characteristics through modification [20]. To solve these problems, different types of fiber hydrogels that mimic the ECM are being studied [8].

Molecular assembly is the method most commonly used for preparing fiber hydrogels. The first step is to assemble the molecules into 1–30 nm fibrils. Following fibril formation, the fibrils can create a single uniform hydrogel through entanglement [21,22]; they can also bundle to form fibers, which subsequently branch and/or entangle to form a hydrogel [23,24,25,26]. However, the structure of a fiber hydrogel prepared in this way is too uniform to imitate the hierarchical structure of the ECM [8,19].

In this paper, we sought to better simulate the hierarchical structure of the ECM using a simple and effective method to prepare a fiber hydrogel. The specific experimental process is shown in Figure 1. Firstly, chitosan microfibers (CMFs) with a size of 1 to 20 μm were prepared using electrostatic self‑organization technology. Then, the CMFs were added to a carboxymethyl chitosan solution (CMCS) to obtain a chitosan fiber suspension. Lastly, this suspension was mixed with oxidized sodium alginate (OSA) solution in equal proportions to obtain an oxidized sodium alginate–chitosan fiber hydrogel (OSA‑CMCS‑CMF). The inhomogeneity of the CMF diameter results in OSA‑CMCS‑CMF presenting a structure highly similar to that of the natural ECM in terms of uniformity. 

In addition, the reversibility and H^+^ sensitivity of the Schiff base reaction confer pH responsiveness [27] and dynamic self‑healing characteristics to OSA‑CMCS‑CMF. In addition, as the most widely used medical polymer materials, sodium alginate and chitosan are not only cheap and easy to obtain but also have the advantages of good biocompatibility and low toxicity. Using OSA‑CMCS‑CMF in cell culture allowed cells to not only grow normally on the hydrogel but also migrate into the gel and gather near the fibers. Rat skin wound repair experiments demonstrated that OSA‑CMCS‑CMF effectively promotes skin wound healing. In short, the structural similarity and good biocompatibility of OSA‑CMCS‑OSA with respect to the ECM make it a good candidate for prospective applications in the field of tissue engineering.

## 2. Materials and Methods

### 2.1. Materials

Sodium alginate (SA) (99%) was purchased from J&K Chemicals (Beijing, China). Carboxymethyl chitosan (CMCS) and chitosan (CM) were purchased from Aladdin Industrial Corporation (Shanghai, China). Polyacrylic acid (PAA), Fehling’s solution, and sodium periodate (NaIO_4_) were purchased from Sigma‑Aldrich (St. Louis, MO, USA). Epidermal growth factor (EGF) was purchased from Beijing Biosynthesis Biotechnology Co. Ltd. (Beijing, China). Fetal bovine serum (FBS), phosphate buffer solution (PBS, pH 7.4), Dulbecco’s modified Eagle’s medium (DMEM), and 1% penicillin‑streptomycin solution were purchased from Corning (Corning, NY, USA). All other reagents and solutions were of reagent grade and used as received.

### 2.2. Methods

#### 2.2.1. Alginate Oxidation

First, 10 g of sodium alginate was added to 50 mL of absolute ethanol to obtain a suspension. Distilled water was used to prepare 50 mL of 0.03 M NaIO_4_ aqueous solution. The NaIO_4_ aqueous solution and sodium alginate suspension were mixed and reacted magnetically for 6 h at a reaction temperature of 25 °C. An equimolar amount of *o*‑diol and sodium periodate was added to terminate the reaction, and 500 mL of absolute ethanol was then used to wash the resultant solution before filtering with suction. After dissolving the OSA, it was dialyzed for 3 days using a dialysis bag with a molecular weight cutoff of 4500 M; OSA was then obtained via freeze‑drying. The reaction is shown in Figure 2a (step 1). The molar ratio of SA to NaIO_4_ was 1:0.6. The degree of oxidation of OSA was about 60%. Since the aldehyde group peak in OSA could not be clearly seen in the Fourier transform infrared (FTIR), we used Fehling’s solution to detect the aldehyde group in OSA. The prepared Fehling’s reagent was added to the OSA solution (2 mL, 10% *w*/*v*) in a boiling water bath, with a brick-red precipitate appearing after 40 min.

#### 2.2.2. Preparation of Fibers

A 0.137 M adipic acid solution was used as the solvent to prepare a 0.01% CM solution, and distilled water was used as the solvent to prepare a 1% PAA solution, resulting in a viscous liquid with a content of 30% and an average molecular weight of 3000 M. The CM solution was slowly added dropwise to the PAA solution until a final volume ratio of 1:3 was reached; the solution was then left to stand at 25 °C for 1 h. Upon completion of the reaction, the solution was frozen at −80 °C, thawed, centrifuged, and freeze‑dried to obtain the CMFs. The reaction is shown in Figure 2a (step 2).

#### 2.2.3. Preparation of Basic Hydrogel

Using 0.9% NaCl as the solvent, a 10% OSA solution and 2.5% CMCS solution were prepared, which were subsequently mixed to generate OSA‑CMCS solutions with different volume ratios. The gelation reaction temperature was 37 °C. A stopwatch was activated to begin timing when the OSA and CMCS solutions were evenly mixed. The state of the hydrogel was determined by repeatedly inverting the sample bottle. When the gel did not reflux upon vial inversion, this indicated that the hydrogel changed from a sol state to a gel state, and the time was immediately recorded from the stopwatch as the gelation time. The reaction is shown in Figure 2a (step 3).

#### 2.2.4. Preparation of Hierarchical Fiber Hydrogel

A 0.9% NaCl solution was used to prepare a 10% OSA solution and CMCS‑CMF solutions containing various proportions of CMF ranging from 0–0.5%. The mixture was reacted at a volume ratio of 1:2 to obtain OSA‑CMCS‑CMF. The gelation reaction temperature was 37 °C. A stopwatch was used to record the gelation time from when the OSA and CMCS‑CMF solutions were evenly mixed. The gelation time was subsequently obtained using the procedure outlined in Section 2.2.3.

#### 2.2.5. Fourier Transform Infrared Spectroscopy

Equal volumes of CMCS, OSA, and OSA‑CMCS solutions were separately added to compressed KBr. After drying in an oven at 37 °C for 24 h, infrared spectroscopy was performed on the above solutions using a microscope infrared spectrometer (spotlight 200i, PerkinElmer, Waltham, MA, USA).

#### 2.2.6. Characterization of CMF Morphology and Surface Element Analysis

Environmental scanning electron microscopy (ESEM, Nova 200, American FEI, Hillsboro, OR, USA) and transmission electron microscopy (TEM, Tecnai G2 20S-TWIN, American FEI, Hillsboro, OR, USA) were used to characterize the morphology of freeze‑dried fibers. The preparation of CMF was achieved through the layer‑by‑layer self-assembly of CM and PAA. In order to explore whether the outermost layer of the prepared CMF was CM or PAA, X‑ray photoelectron spectroscopy (XPS, ESCALAB250Xi, Thermo Fisher Scientific‑CN, Shanghai, China) was used. Thus, 50 µg of CMF was evenly applied to a conductive tape for XPS scanning detection.

#### 2.2.7. Environmental Scanning Electron Microscopy

The hydrogel and the back connective tissue of rats were observed using an environmental scanning electron microscope (ESEM) with normal water content. Before observation, the rat back connective tissue or hydrogel was maintained at −20 °C for 2 h on a refrigerating table, followed by placing the sample on a microscope stage in a vacuum chamber for observation.

#### 2.2.8. Shear Force During Injection

Rheology (MCR 72/92, Anton Paar Shanghai, Trading Company, Shangai, China) is used to measure the change in the shear force of the hydrogel in the first 150 s during the gelation. Mix 0.2 mL of 10% OSA with 2.5% CMCS (0.1% CMF) by 1/2 and then quickly place it on the operating table of the rheometer to measure the shear force.

#### 2.2.9. Degradation Kinetics of Hydrogel

The degradation laboratory is carried out at room temperature. The appropriate amount of OSA‑CMCS‑CMF was sealed in a centrifuge tube and stored at room temperature. Weigh the mass of the hydrogel with a balance once a day. Observe the changes in the hydrogel.

#### 2.2.10. Self‑Healing of Hydrogel

Use a blade to cut the hydrogel into two pieces, and then stick the two pieces of hydrogel together. Finally, put the hydrogel in a 37 °C thermostat, and observe the state of the two pieces of hydrogel.

#### 2.2.11. In Vitro Cell Culture

A 30 μg/mL epidermal recombinant growth factor (EGF) solution was prepared using 0.9% NaCl as a solvent, and a 10% OSA solution and a CMCS solution containing 0.1% CMF were then prepared using the EGF solution as a solvent followed by sterilization in a UV cabinet for 3 days. The sterilized solutions were mixed and reacted in a 96‑well culture plate to obtain EGF‑OSA‑CMCS‑CMF. Finally, a suspension of epidermal fibroblasts (L929, National Science & Technology Infrastructure, Shanghai, China) was added to the 96‑well plate, and the L929 cells were cultured in OSA‑CMCS and OSA‑CMCS‑CMF in a cell incubator according to the above experimental methods.

The medium was aspirated and replaced after culturing for both 1 and 3 days. Cell Counting Kit‑8 (CCK8, Beyotime Biotechnology, Shanghai, China) reagent was added before continuing with culturing for 2 h. The microplate reader (MR-96A, The Company of Mindray Medical Devices, Shenzhen, China) is used to measure the relationship between absorbance and optical density (OD). Taking the cells cultured on OSA‑CMCS‑CMF as the research object, the medium in the well plate was aspirated using a pipette and washed three times with PBS. The configured acridine orange/ethidium bromide (AO/EB) staining solution was added, and the solution was left to stand for 10 min in the dark. The stained solution was aspirated, washed twice with PBS, and observed by fluorescence microscopy (DYF‑850, The Company of DIANYING Optical Instrument, Shanghai, China) using a 20× objective lens. After staining, the dead and living cells were appeared red and green, respectively, under the excitation of different corresponding wavelengths of monochromatic light.

#### 2.2.12. Animal Experiments

Sprague‑Dawley (SD) male rats (300–350 g) were selected for animal experiments. Four defect areas with diameters of 10 mm were symmetrically generated on the back epidermal layers of the rats. Equal amounts of OSA‑CMCS, OSA‑CMCS‑CMF, and EGF‑OSA‑CMCS‑CMF were affixed to the defects to create different experimental groups. To prevent the rats from licking the wounds and gel, gauze was used as a secondary dressing to protect the wound. The last experimental group was a blank control with no hydrogel dressing. The back wounds of all experimental rats were photographed after 3, 5, 9, and 14 days, and the wound area was evaluated using ImageJ software (ImageJ, 1.8.0, National Institutes of Health, Bethesda, MD, USA). The rats were sacrificed at 14 days; the skin area which underwent wound healing was then cut off and fixed with 4% glutaraldehyde prior to staining with hematoxylin and eosin (H&E) and Masson’s trichrome. All animal experiments were approved by the Institutional Animal Care and Use Committee of the National Center for Nanoscience and Technology (License number: NCNST21‑2010‑07)

## 3. Results

### 3.1. Morphological Characterization and Chemical Composition

Before the preparation of the hierarchical hydrogel, a basic hydrogel (CMCS‑OSA) was first prepared via a Schiff base reaction between OSA and CMCS [28]. Since the –C=O peak could not be clearly identified in the infrared spectrum of OSA, Fehling’s solution was used to detect the aldehyde group in OSA, as shown in Figure 2(b-1, b-2). After adding Fehling’s reagent to the hydrogel in a boiling water bath for 20 min, a brick-red precipitate could be clearly seen at the bottom of the bottle, indicating that sodium alginate (SA) was successfully oxidized with the formation of aldehyde groups. Then, Fourier transform infrared spectroscopy was used to illustrate the chemical reactions involved in the preparation of hydrogels, as shown in Figure 2(b-3). Compared with OSA and CMCS, OSA‑CMCS presented a new absorption peak at 1680 cm^−1^, corresponding to a tensile vibration peak belonging to –C=N. The appearance of this peak indicated the formation of Schiff base bonds.

In order to imitate hierarchical structures, chitosan microfibers (CMF) were prepared via electrostatic self-assembly of chitosan (CS) and polyacrylic acid (PAA) [29]. Environmental scanning electron microscopy (ESEM) showed that the diameter of the prepared CMF ranged from 1 to 20 µm (Figure 2c). We speculate that this was caused by the different growth cycles of different fibers during CMF self‑assembly. Transmission electron microscopy (TEM) images indicate the filamentous fiber properties of the CMF. It can be seen that the density of the fiber is uniform. XPS elemental analysis (Figure 2g) showed that nitrogen was present on the surface of the CMF, belonging to the amino group of the chitosan molecule. This result indicates that there were exposed amino groups on the surface of the CMF, which could undergo Schiff base reactions with OSA in the experiment.

Then, the oxidized sodium alginate‑chitosan fiber hydrogel (OSA‑CMCS‑CMF) was prepared via Schiff base reaction. CMFs were first added to the CMCS solution and then mixed with the OSA solution to achieve gelation. During gelation, CMFs were anchored to the OSA chain via Schiff base reaction. Finally, CMFs and OSA‑CMCS together formed a hierarchical hydrogel with fibers. Figure 2d shows that OSA‑CMCS is a hydrogel with a uniform porous structure. However, upon adding CMFs to OSA‑CMCS, the hydrogel changed to a hierarchical structure, as shown in Figure 2e. These fibers, with a diameter ranging from a few micrometers to 10 µm, formed a typical hierarchical structure with the base hydrogel.

Next, we observed the morphology and structure of loose connective tissue (LCT) on rat backs, as shown in Figure 2f. It can be seen that LCT was composed of abundant fibers and a gel matrix. The similarity in morphology and structure of LCT and OSA‑CMCS‑CMF, especially in terms of fiber size, was confirmed through comparison. Normal distribution analysis of the fiber diameters inside OSA‑CMCS‑CMF and LCT was applied, once again highlighting their similarity (Figure 3h).

### 3.2. Gelation Time

As this hydrogel is intended for injectable use, the gelation time must be precisely controlled. First, the effect of the volume ratio of OSA/CMCS on the gelation time was investigated, as shown in Figure 3a. In the range of 0.5–1.25, as the ratio of CMCS to OSA increased, the gelation time of the hydrogel gradually decreased due to an increase in the concentration of amino groups in the solution, thereby increasing the rate of the Schiff base reaction and ultimately accelerating hydrogel formation. At a ratio of 1.25, the relative concentration of amino and aldehyde groups resulted in the best chemical reaction rate and, thus, the shortest gelation time. When the volume ratio exceeded 1.25, any further increase reduced the concentration of aldehyde groups, resulting in a decrease in the chemical reaction rate and an increase in gelation time.

In addition, the effect of CMF concentration on gelation time was evaluated, as shown in Figure 3b. When the CMF concentration was less than 0.1%, it had little effect on the gelation time. However, beyond this point, the gelation time increased with the increase in CMF concentration, perhaps due to the amino groups on the excess CMF competing with those on the CMCS molecular chains in the solution, eventually prolonging the gelation time. In addition, as the CMF content increased, the turbidity of the hydrogel increased. Whether the gel is used as a wound dressing or an in situ injection, the gelation time must be controlled within an appropriate range [30]. Furthermore, in wound dressing applications, hydrogels with higher transparency are preferred as they facilitate the monitoring of wound condition [28]. Therefore, we selected a 10% OSA solution and 2.5% CMCS (containing 0.1% CMF) solution in a 1:2 ratio to prepare fiber hydrogels for subsequent cell and animal experiments.

### 3.3. The Properties of OSA‑CMCS‑CMF

The shear force during injection, degradation time, and self‑healing properties of the hydrogel were explored as a function of the selected concentration.

#### 3.3.1. Shear Force Properties

In order to illustrate the good injection performance of the hydrogel, the shear force of OSA‑CMCS‑CMF during gel reaction was measured, as shown in the Figure 3c. When the two solutions are just mixed, the shear force of the hydrogel is extremely small. At this time, the hydrogel can be injected through a 2.5 mL syringe, as shown in Figure 3e. With the progress of the gelation reaction, the shear force of the hydrogel rapidly increased after 90 s. At this time, the degree of gelation is higher and injection becomes difficult.

#### 3.3.2. Degradation Time Properties

The degradation rate of the hydrogel within 7 days is counted in order to explore the degradation of the OSA‑CMCS‑CMF, as shown in Figure 3d. It can be seen from the figure that as the number of days increases, the quality of the hydrogel decreases slowly from day 1 to day 7, and the degradation rate of the hydrogel gradually decreases. Figure 3f shows the state of the hydrogel after gelation for 5 min and gelation for 7 days. It can be seen that the color of the hydrogel changes from transparent to orange with time, which indicates that the hydrogel is slowly oxidized and degraded. After in vitro observation, it is found that the degradation time ranged from 14 to 21 days. The in vitro degradation experiments are shown in Figure 3g.

The degradation of OSA‑CMCS‑CMF occurred through a breakdown of polysaccharides, so its degradation products are non-toxic. This rapid and nontoxic degradation has major advantages in applications such as tissue repair.

#### 3.3.3. Self‑Healing Properties

It is worth mentioning that due to the reversibility of the Schiff base reaction and the pH (H^+^) sensitivity [31], OSA‑CMCS‑CMF also has dynamic response capabilities which change with the pH value [27,32]. Because the Schiff base reaction conditions were mild and reversible, the hydrogel exhibited good dynamic self‑healing properties, making it advantageous for applications in the field of tissue engineering. As shown in Figure 3h, we cut the hydrogel into two halves and placed the halves next to each other. After maintaining a temperature of 37 °C for 4 h, the hydrogel pieces were automatically reconstituted into one piece.

### 3.4. Cytocompatibility of OSA‑CMCS‑CMF

Next, the cytocompatibility of the hydrogel was explored. L929 fibroblasts were cultured on the surface of the hydrogel, and the cell viability was measured using the CCK8 method, as shown in Figure 4a. After 1 day of culture, the activity of cells grown on OSA‑CMCS‑CMF exceeded that of cells grown on OSA‑CMCS, with this phenomenon becoming more marked after 3 days. This indicates that OSA‑CMCS‑CMF can promote the growth and proliferation of L929 cells. We also explored the effects of epidermal growth factor (EGF) on cell viability. The results show that the addition of EGF enhanced the effect of OSA‑CMCS‑CMF on fibroblast proliferation. These results indicate that OSA‑CMCS‑CMF has good cytocompatibility. More interestingly, L929 fibroblasts cultured on the surface of OSA‑CMCS‑CMF were found inside OSA‑CMCS‑CMF after a period of culture. We saw a large number of spherical particles gathered on the fibers in the cross-section of the hydrogel in which L929 fibroblasts were cultured (Figure 4b). Upon comparing the diameter of these particles with the diameter of L929 fibroblasts in the hydrogel under an optical microscope (Figure 4c), it was determined that these particles were L929 fibroblasts. Fluorescence staining was performed on these cells (Figure 4d), and the results show that they were all living. This indicated that cells could migrate into the hydrogel during growth and proliferation, leading to aggregation and further growth on the inner fibers. This phenomenon is very similar to the adherence of cells to the fiber surface in the ECM, enabling signal transmission through attachment points [8].

In summary, OSA‑CMCS‑CMF had a more distinct promotion effect on cell growth and proliferation than OSA‑CMCS. We attribute this to the fiber network structure and its good cytocompatibility.

### 3.5. Skin Wound Repair

Following the above findings, we used OSA‑CMCS‑CMF in rat skin wound repair experiments to explore the feasibility of the material as an injectable hydrogel for in vivo tissue repair. Figure 5a shows that the wound defects in the blank control and experimental groups were both repaired after 14 days. However, the healing effect with OSA‑CMCS‑CMF was better than with the blank and OSA‑CMCS. The addition of EGF further promoted wound healing. As shown in Figure 5b, the changes in wound healing rate across different experimental groups could be intuitively observed. The healing rates of wounds treated with OSA‑CMCS‑CMF and EGF‑OSA‑CMCS‑CMF were higher than those of the blank group and OSA‑CMCS, with EGF‑OSA‑CMCS‑CMF showing the highest rate of wound healing.

We further evaluated the wound healing effect using histological analysis. Figure 5c shows hematoxylin and eosin (H&E) and Masson’s trichrome staining performed after the healing of defective tissue. Layers treated with the OSA‑CMCS‑CMF and EGF‑OSA‑CMCS‑CMF were completely covered by the epidermis. In comparison, the group which underwent treatment with OSA‑CMCS and the blank control group showed incomplete and uneven growth. In the OSA–CMCS–CMF and EGF‑OSA‑CMCS‑CMF groups, substantially more granulation tissues were observed in the subepidermal layers than in the blank and OSA‑CMCS groups. The granulation tissue of the EGF‑OSA‑CMCS‑CMF group completely filled the defect, with no distinct difference from the surrounding normal tissue. Newborn hair follicles were also apparent in this section. Substantially more hair follicles were found in the OSA‑CMCS‑CMF and EGF‑OSA‑CMCS‑CMF groups than in the blank and OSA‑CMCS groups. The Masson’s trichrome-stained sections showed that compared with the blank and OSA‑CMCS groups, more collagen fibers were present in the wounds treated with the OSA‑CMCS‑CMF and EGF‑OSA‑CMCS‑CMF, and their arrangement was more compact and orderly.

## 4. Discussion

In this study, a microfiber with diameter ranging 1–20 µm was prepared via electrostatic self‑assembly. XPS characterization showed that the outermost layer of this fiber was chitosan, containing amino groups which could be used as the linking site of the hydrogel. This fiber was connected with OSA and CMCS through Schiff base bonds to prepare a hierarchical fiber hydrogel (OSA‑CMCS‑CMF). Upon comparison with the microstructure of the rat back connective tissue, it was found that OSA‑CMCS‑CMF had a similar morphology and structure. Through a statistical analysis of the internal fiber diameter, it was found that the size distribution range of internal fibers for OSA‑CMCS–CMF and the ECM were very similar, confirming the proposed hierarchical fiber hydrogel has a bionic structure.

The CCK8 and dead/live staining experiments showed that the OSA‑CMCS‑CMF has good cytocompatibility. During the investigation process, it was found that L929 cells migrated to fibers inside OSA‑CMCS‑CMF, similar to the cell adherence to and growth on fibers inside the ECM [8]. We speculate that the similar fiber size in OSA‑CMCS‑CMF and the ECM provides an adequate adhesion site for cells, stimulating their migration and growth. This phenomenon shows that the hierarchical fiber hydrogel prepared by imitating the ECM structure is successful. At the same time, this is also the advantage of the hierarchical fiber hydrogel, which allows cells to find micron-scale adhesion sites in the three-dimensional system, so that they can sense external forces like in ECM [9]. Finally, the rat skin wound repair experiments demonstrate that this hierarchical fiber hydrogel can promote the healing of skin wounds. We speculate that during the healing process of wounds coated with hierarchical fiber hydrogel, some fibroblasts in the wound area will migrate to the inside of the hydrogel to accelerate wound healing. The cell and animal experiments show that this biomimetic hierarchical fiber hydrogel not only has good biocompatibility but also promotes the repair of epidermal wounds.

## 5. Conclusions

In summary, the hierarchical biomimetic hydrogel prepared using a simple strategy not only successful imitated the fiber structure of the ECM, but also showed excellent biocompatibility and wound healing promotion effects in actual cells and wound dressings. This study showed that the prepared OSA‑CMCS‑CMF has excellent biocompatibility, good injectability, self‑repair ability, and good degradation characteristics in terms of being rapid and nontoxic. These advantages not only allow it to be used as a skin wound dressing but also in the repair of soft tissue as a function of its injectable properties and ability to carry living cells. In addition, it can be used as a carrier for drugs and other substances, transport medium, and support material in the novel method of fascial space injection [33]. This hydrogel thus has broad potential applicability as a tissue engineering material.

## Figures and Tables

**Figure 1 materials-13-05277-f001:**
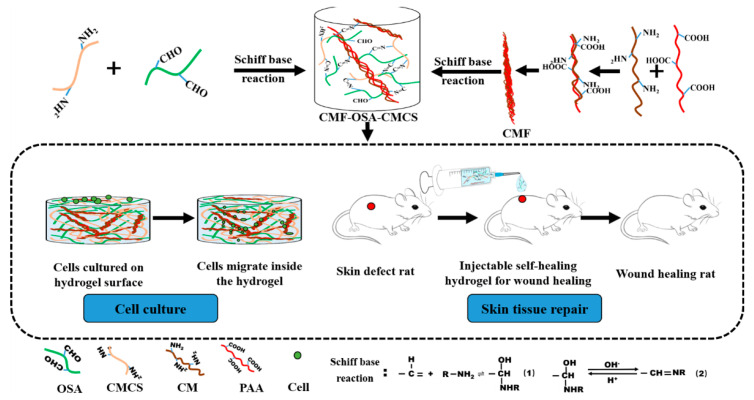
Design flow chart for preparing fiber mesh hydrogel and schematic of cell animal experiment. OSA, oxidized sodium alginate; CMCS, carboxymethyl chitosan solution; CM, chitosan; PAA, polyacrylic acid.

**Figure 2 materials-13-05277-f002:**
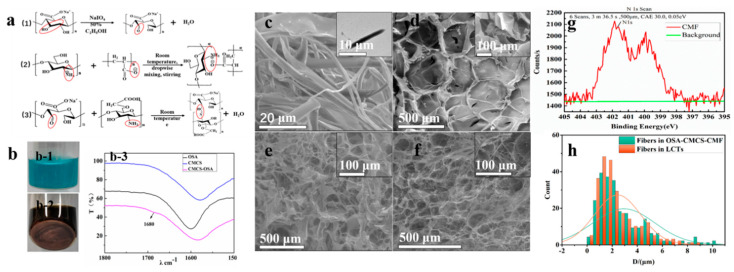
(**a**) Reactions and principles designed in the article: (1) Schiff base reaction equation of OSA and CMCS; (2) schematic diagram of prepared microfiber (CMF) preparation via electrostatic self-assembly of CM and PAA; (3) sodium alginate oxidation. (**b**) Identification of chemical groups: (**b-1**) Fehling’s solution and oxidized sodium alginate mixture; (**b-2**) solution of b1 was heated for 40 min, (**b-3**) Fourier transform infrared spectroscopy of OSA, CMCS, and OSA‑CMCS. Environmental scanning electron microscopy (ESEM) images of (**c**) prepared microfiber (CMF), with TEM of a single CMF (inset, upper right corner), (**d**) OSA‑CMCS topographic structure, (**e**) OSA‑CMCS‑CMF topographic structure, and (**f**) connective tissue under the skin of rat backs. (**g**) X‑ray photoelectron spectroscopy (XPS) element analysis. (**h**) Normal distribution of fibers in the loose connective tissue of rat backs (LCT) and of CMFs in OSA‑CMCS‑CMF.

**Figure 3 materials-13-05277-f003:**
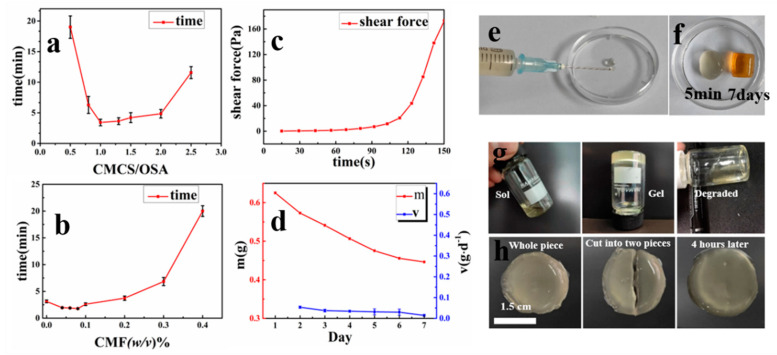
(**a**) Different CMCS/OSA ratios and gelation times. (**b**) Gelation time diagram of CMF with different concentrations of CMCS. (**c**) The shear force of OSA‑CMCS‑CMF in 0–150 s during gel reaction. (**d**) The degradation of OSA‑CMCS‑CMF hydrogel within 7 days at room temperature (m: the quality of the hydrogel, V: the reduced quality of the hydrogel per day). (**e**) The injectability of the hydrogel. (**f**) The state of OSA‑CMCS‑CMF after gelation for 5 min and 7 days. (**g**) Hydrogel degradation map (from left to right: sol, gel, and degraded state). (**h**) Hydrogel self‑healing effect chart.

**Figure 4 materials-13-05277-f004:**
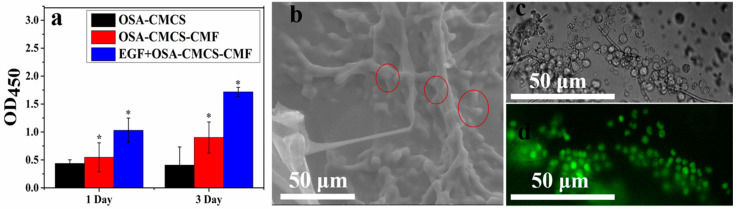
(**a**) Cell growth on different hydrogels after 1 and 3 days of culture (Comparing the data of each group with the OD value of the OSA–CMCS group on the first day, * *p* < 0.05, there is a significant difference. (**b**) ESEM of cells growing inside the hydrogel. (**c**) Light micrograph of cells growing inside the hydrogel. (**d**) Fluorescence staining of cells growing inside the hydrogel.

**Figure 5 materials-13-05277-f005:**
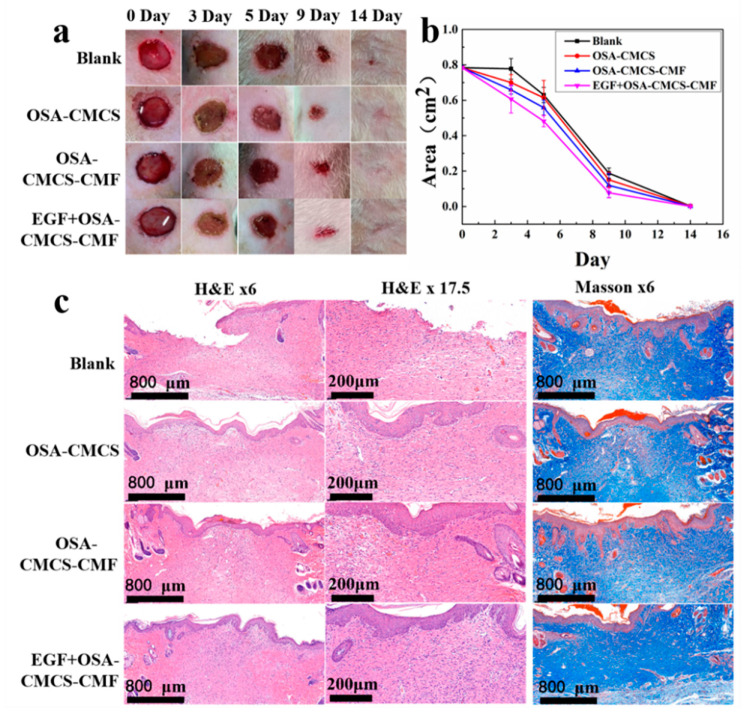
(**a**) Photos of the blank control, OSA‑CMCS, OSA‑CMCS‑CMF, and EGF‑OSA‑CMCS‑CMF treatments showing the repair of back skin defects in rats after 0, 3, 5, 9, and 14 days. (**b**) Statistical maps of the change in defect area during the wound healing process. (**c**) Images of wound pathological staining after 14 days.

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
