# Peer review of "A Self-Healing Hierarchical Fiber Hydrogel That Mimics ECM Structure"

_materials, 2020, doi:10.3390/ma13225277_

Round 1

Reviewer 1 Report

The original article “A self-healing hierarchical fiber hydrogel for ECM mimicking”  presented by Kai Li et al., demonstrated that an hybrid hydrogel based on alginate-cellulose-chitosan exhibits some features of interest to be used as wound dressing. One of the key characteristics is the architectural organization of this microfibrous hydrogel which shows some similarity to the natural ECM of skin rat tissue.

This works has a certain degree of novelty and this hydrogel might be an interesting candidate for such healthcare application.

Nevertheless, I have some concerns and suggestions for improvement.

  • In general, this manuscript contains numerous English mistakes (e.g. minicking in the title) which must be corrected. Numerous sentences are not clear and must be reformulated. I would highly recommend to have this article corrected by English native speaker as it is not satisfying at this stage.
  • I would recommend to revisit the keywords, as for instance “bionics” does not appear anywhere else in the text than in this list.
  • Please, insure that before using acronyms, the full names of the components are first written.
  • The authors must distinguish in the terminology within the manuscript when biomaterials assessment are done in vitro with cells (=cytocompatibility) to investigations conducted in the body (biocompatibility).
  • Some important statements -claims in the introduction are missing supportive literatures.
  • The M&M is not complete and many equipment and characterizations appearing in the result section are actually not mentioned anywhere in the methodology. The information about the brand of the devices used and origin are also missing. In this section, many units are missing as well. Some chapter´s heads would deserve rephrasing as the meaning might be confusing - or incorrect.
  • The in vivo Methodo is insufficiently described. Ethic agreement, n number etc… are missing.
  • Please, describe methodology for gelation experiment. It would be important to do it at skin temperature (35-37ºC). Same for degradation study. Could the authors provide a quantification of the kinetic of degradation for all the groups?
  • I would re-work on all the graphs to improve their resolution. It could be worth to fully write the meaning of each acronym at least in the caption of the figure 1
  • Check that the figure number appearing in the manuscript is correct.
  • Statistics are missing in some graphs to state that something is improved or not.
  • Fig 4: Could the authors comment on the shape of the fibroblasts growing into the hydrogel. The cells do not really exhibit a “normal-like” phenotype with spindle shape cytoplasm, filopodia.. but rather a round-shape phenotype usually seen when cells are more in a non-proliferative/ non-adhering behavior….which contradicts the message delivered for this figure.

  • Conclusion must be re-organized as it brings some discussion points and a proper discussion section is missing.

Reviewer 2 Report

The manuscript by Li et al. describes the preparation of fiber-based hydrogels to mimic the hierarchical structure of native ECM. Though the mechanism of hydrogel formation sounds interesting, there are some experiments missing. Also, it is not clear from the paper what is the advantage of the fiber-based structure. Therefore the manuscript needs some revisions before publication. Please find below for specific comments:

  1. Abstract and introduction need some improvements.
  2. The degree of oxidation of alginate should be reported.
  3. Figure 2 is blurry and the FTIR peaks discussed in the text are not visible in the spectra.
  4. The authors showed a picture of a degraded hydrogel; however it is not described how the degradation was performed.
  5. In section 3.3 the authors claim that the hydrogels were biocompatible; however what is described in that section is the viability of cells seeded onto the hydrogels and therefore the cytocompatibility.
  6. In the conclusion, the authors state that the hydrogels presented in the manuscript have “good injectability”. There are no data supporting this statement. Rheological data should be added to show whether the hydrogels are injectable.

Round 2

Reviewer 2 Report

The manuscript results quite improved after revisions and I recommend to be accepted in the present form.